# Effects of Membrane and Biological Target on the Structural and Allosteric Properties of Recoverin: A Computational Approach

**DOI:** 10.3390/ijms20205009

**Published:** 2019-10-10

**Authors:** Alberto Borsatto, Valerio Marino, Gianfranco Abrusci, Gianluca Lattanzi, Daniele Dell’Orco

**Affiliations:** 1Department of Neuroscience, Biomedicine and Movement Sciences, Section of Biological Chemistry, University of Verona, 37134 Verona, Italy; 2Department of Physics, University of Trento, 38123 Trento, Italy; 3Department of Translational Research and New Technologies in Medicine and Surgery, University of Pisa, 56026 Pisa, Italy; 4Trento Institute for Fundamental Physics and Applications (INFN-TIFPA), Via Sommarive 14, Povo, 38123 Trento, Italy

**Keywords:** recoverin, myristoyl switch, molecular dynamics, phototransduction, rhodopsin kinase, protein structure networks

## Abstract

Recoverin (Rec) is a prototypical calcium sensor protein primarily expressed in the vertebrate retina. The binding of two Ca^2+^ ions to the functional EF-hand motifs induces the extrusion of a myristoyl group that increases the affinity of Rec for the membrane and leads to the formation of a complex with rhodopsin kinase (GRK1). Here, unbiased all-atom molecular dynamics simulations were performed to monitor the spontaneous insertion of the myristoyl group into a model multicomponent biological membrane for both isolated Rec and for its complex with a peptide from the GRK1 target. It was found that the functional membrane anchoring of the myristoyl group is triggered by persistent electrostatic protein-membrane interactions. In particular, salt bridges between Arg43, Arg46 and polar heads of phosphatidylserine lipids are necessary to enhance the myristoyl hydrophobic packing in the Rec-GRK1 assembly. The long-distance communication between Ca^2+^-binding EF-hands and residues at the interface with GRK1 is significantly influenced by the presence of the membrane, which leads to dramatic changes in the connectivity of amino acids mediating the highest number of persistent interactions (hubs). In conclusion, specific membrane composition and allosteric interactions are both necessary for the correct assembly and dynamics of functional Rec-GRK1 complex.

## 1. Introduction

The small (23 kDa) Ca^2+^-binding protein recoverin (Rec) is a prototypical member of the neuronal calcium sensor (NCS) family mostly expressed in vertebrate photoreceptors, where it plays a role in the regulation of the phototransduction cascade [1,2]. Light triggers a transient drop in the intracellular concentration of Ca^2+^ and such change is promptly detected by a group of calcium sensors including Rec, which reversibly bind to specific targets in a Ca^2+^-dependent manner. This specific binding permits the complex regulation of the signaling cascade and is fundamental to bring the cell back to the dark state, where light sensitivity is maximal [3].

As in many other NCS proteins, the N-terminus of Rec is acylated, and the most abundant post-translational modification found in vivo is myristoylation [4]. Rec has two functional EF-hands, namely EF2 and EF3 (Figure 1), which allow the binding of two Ca^2+^ ions in the dark. The binding process triggers a structural transition from a “tense” (Rec-T), compact conformation to a “relaxed” (Rec-R), more extended conformation. In the Rec-T conformation, the myristoyl group is sequestered in a hydrophobic pocket, while in Rec-R the myristoyl moiety is solvent-exposed [5,6]. The exposure of the myristic moiety is concomitant to the exposure of hydrophobic residues on the protein surface, thus providing an anchor for binding Rec to the surface of photoreceptor disc membranes [5,7]. This, in turn, allows the binding with the rhodopsin kinase (Rec-GRK1) and the Ca^2+^-dependent regulation of its activity [8,9]. The myristoyl switch mechanism constitutes a functional hallmark of Rec as well as other NCS proteins including NCS-1 [10] and VILIP-3 [11]. The peculiar physicochemical details have been long investigated [6,12,13].

Recently, the binding of recoverin to a biological membrane was simulated by molecular dynamics (MD) in an elegant study by Timr et al. [14]. The extensive all-atom and coarse-grained simulations suggested that the disordered C-terminal segment of Rec and/or the presence of the GRK1 target stabilize the conformation of the N-terminal domain, a condition that was found to be necessary for the spontaneous insertion of the myristoyl into the membrane. Moreover, the presence of negatively charged lipids was concluded to be necessary to stabilize the correct orientation of the protein with respect to the lipid bilayer [14]. While the conclusions in that study are in line with a number of experimental results, the lipid composition of the membrane used in Timr et al. (80% phosphatidylcholine (PC) and 20% phosphatidylglycerol (PG)) differs significantly from that of the rod outer segment (ROS) discs. Indeed, the membrane of ROS discs is formed by similar amounts of phosphatidylethanolamine (PE; 42%) and phosphatidylcholine (PC; 45%), while the cationic lipid phosphatidylserine (PS) is present in lower amounts (approximately 14%) [15,16]. In addition, a spatially heterogeneous amount of cholesterol (30% to 5%) decreases both along the axis of the outer segment and with disc ageing [17,18,19], and has important functional consequences, such as the modulation of rhodopsin stability and activity [19] and the formation of lipid rafts affecting Rec and GRK1 activities [20]. The studies of Rec binding to the membrane should therefore account for a realistic membrane composition in order to infer physiological mechanisms. 

NCS proteins exert their function via conformational transitions that follow the binding of specific cations to their multiple EF-hand motifs. The loading state of each EF-hand thus dictates the signaling state of the protein and must be promptly communicated to the amino acids that form the interface with the target. Hence, the precise switch mechanism of any NCS protein requires a long-distance allosteric communication that connects groups of amino acids by noncovalent interactions. In this view, a protein is described as a graph representing a specific protein structure network (PSN) [21], in which amino acids are nodes connected with one another by persistent hydrophobic, electrostatic and/or H-bond interactions, which represent the arches or edges, and define precise intra-molecular routes [22,23]. The PSN paradigm applied to NCS proteins undergoing extensive MD simulations has shown a great potential in disclosing the role of key amino acids in mediating allosteric mechanisms, which is further confirmed by their evolutionary conservation among homologous proteins [24,25].

This study presents exhaustive and unbiased MD simulations of Rec, both uncomplexed and in complex with a peptide from the GRK1 target (Rec-GRK1). In both cases, the spontaneous insertion of the myristoyl moiety into a patch of a membrane with a lipid composition that mimics that of ROS discs was observed. The mechanism of membrane burial of the myristoyl moiety is clearly defined, which requires a specific temporal sequence of salt bridges formed between basic residues of Rec and the anionic head groups of PS lipids. The presence of the GRK1 target affects the network of interactions stabilizing the protein-membrane complex, while the presence of the membrane significantly changes the allosteric communication between EF hands and the target interface in Rec. This study concludes that a fully functional Rec-GRK1 complex can be obtained only as a synergic combination of specific membrane composition and selective allosteric interactions.

## 2. Results and Discussion

Both uncomplexed Rec and Rec in complex with a peptide containing the first 25 amino acids of the GRK1 target (Rec-GRK1) were simulated in all-atom MD simulations that included explicit solvent and a lipid bilayer patch with a composition mimicking that of a ROS disc membrane (Figure 1A,B). The myristic moiety was initially completely solvent-exposed and membrane binding was monitored by following the time evolution of its solvent accessible surface area (SASA). This is expected to drop significantly upon membrane burial. Seven 200 ns MD replicas were run for each different simulated state (Rec-R and Rec-GRK1), two of which started from the protein at 1 nm distance from the membrane bilayer. The following replicas started when the protein (Rec-R or Rec-GRK1) was anchored to the membrane and the insertion process was considered to be complete, as explained below. Full reproducibility of the captured structural dynamics was assessed for each group of replicas, and the replicas of the membrane-bound Rec states were thus concatenated in a single, 1 μs simulation (see Material and Methods).

### 2.1. Spontaneous Insertion of the Myristoyl Moiety of Rec and Rec-GRK1 into the Membrane

All-atom simulations captured two events of spontaneous membrane insertion for each of the simulated Rec states in only one of the two 200 ns replicas, whose temporal evolution is reported in Figure 1C,D. The initial diffusion above the membrane occurred without persistent physical contacts between the protein and the membrane for the first 100 ns in both states. For the uncomplexed Rec-R state, the SASA of the myristic moiety showed lower amplitude oscillations compared to the Rec-GRK1 states in the first 100 ns of simulations, and the membrane insertion started at 141 ns to be fully completed after 23 ns (Figure 1C). The unsuccessful insertion was accompanied by significantly enhanced oscillations in the SASA profile. The myristic moiety in the Rec-GRK1 complex showed overall higher SASA oscillations in both successful and unsuccessful insertion events (Figure 1D). When the insertion process started (t = 153 ns), it was completed significantly faster (14 ns) compared to the uncomplexed state. 

While hydrophobic interactions are crucial for completing the insertion of a peripheral membrane protein into the membrane and govern the burying process, electrostatic interactions between the polar head groups of lipids and basic amino acids contribute significantly in driving the binding process [26]. In particular, the protein macrodipole interacts with the membrane electric field and drives the orientation of the protein with respect to the interface. The experimental evidence supports an important role for the transmembrane potential in driving the interaction between Rec and the membranes, while its orientation on the lipid bilayer was found not to depend on the presence of myristoyl [27]. When projected on the *z*-axis (orthogonal to the membrane plane), the macrodipole of Rec showed important fluctuations and sign reversal (Appendix A) in the MD simulations that did not culminate in membrane insertion. The oscillations were instead significantly lower and no sign reversal was observed in the case of successful membrane insertion (Appendix A), indicating the importance of the correct orientation of the protein in the binding process. A dynamic representation of the spontaneous insertion of Rec-R into the membrane reporting the orientation of the protein macrodipole is shown in Appendix A

### 2.2. Binding of Rec to the Membrane is a Multi-Step Process Involving Specific Electrostatic Interactions

In spite of their relatively low abundance (20%) compared to the other lipid types (40% PE; 40% PC), PS lipids were found to be crucial for initiating and driving the membrane binding process of both Rec-R and Rec-GRK1. The snapshots of the early steps of membrane insertion of myristoylated Rec-GRK1 are shown in Figure 2. 

In both simulations resulting in successful membrane insertion, the N-terminal domain of Rec kept its initial orientation towards the membrane, in agreement with previous MD simulations [14]. The methyl end of the myristoyl chain initially entered into a transient gap formed between the head groups of lipids, before penetrating the membrane to be eventually accommodated among the fatty-acid chains of the surrounding phospholipids. This process occurred via multiple steps involving the electrostatic interactions between specific amino acids at the N-terminal domain of Rec and lipid polar heads. For Rec-GRK1, at t = 153 ns a first persistent salt bridge was established between R43 and the polar head of a PS lipid (Figure 2A,D). This initial interaction was necessary to ensure the correct orientation of the myristoyl group, whose membrane insertion started at 157 ns (Figure 2B,E). In order to stabilize the orientation of the Rec-GRK1 complex with respect to the membrane and complete the myristoyl anchoring, a second salt bridge between R46 and the polar head of another PS lipid was established at t = 167 ns (Figure 2C,F). A very similar multi-step mechanism was observed for Rec-R, which highlighted the importance of the anionic head groups of PS lipid in initiating the membrane binding process.

### 2.3. A Specific Network of Electrostatic Interactions Stabilizes the Membrane-Bound State of Rec-R and Rec-GRK1

Once the first contacts were established, both Rec-R and Rec-GRK1 remained peripherally bound to the membrane with the myristic moiety completely buried for the rest of the simulation (1 μs), indicating a stable interaction. Moreover, the proteins maintained their initial orientation, with the principal axis oriented at approximately 45 degrees with respect to the membrane normal, consistent with what was observed by solid-state Nuclear Magnetic Resonance (NMR) spectroscopy [28]. In that work, the authors found that a number of basic residues in Rec’s N-terminal domain, namely K5, K11, K37, R43 and K84, were interacting with the membrane thus being ultimately responsible for keeping such stable orientation with respect to the membrane. This orientation guaranteed an unhindered exposure of the hydrophobic crevice formed upon extrusion of the myristic moiety and thus an ideal interface to accommodate the GRK1 interacting target [28]. Since the membrane composition used in the NMR experiments was formed by 80% PC and 20% PG, and the same composition was used in the MD simulations by Timr et al. [14], this study sought to extend the analysis of protein-lipid contacts to a more realistic membrane with a ROS disc-like lipid composition. However, instead of defining the contact between amino acids and lipids by using a simple distance-based criterion, a deeper analysis of the persistence of each possible protein-lipid interaction was performed, distinguishing between H-bonds, electrostatic and hydrophobic interactions, and in fact extending the PSN paradigm to a protein-membrane case. 

Surprisingly, Rec-R and Rec-GRK1 displayed a significantly different pattern of noncovalent protein-membrane interactions (Figure 3). The absence of the GRK1 target resulted in a higher number of persistent protein-lipid interactions compared to the Rec-GRK1 state, and for five residues (N3, S4, K5, R43 and R46), the interactions were mostly H-bonds, persisting over 50% of the simulated timeframe. While H-bonds between polar residues and the lipid head groups were the most abundant interactions, all basic residues forming H-bonds were found to also form very persistent pure electrostatic interactions, as shown especially by K37 and R46. The myristic moiety in the case of Rec-R formed equally persistent H-bonds and hydrophobic interactions (40% of the time frame).

While for the Rec-GRK1 state a lower number of persistent protein-lipid interactions were observed in the time course of 1 μs simulations, the protein-target complex was stabilized by a specific pattern of electrostatic interactions involving the basic residues K5, R43 and R46. All of these persisted for over 50% of the simulation time-frame, and a significantly higher persistence (approximately 70%) of the hydrophobic contacts were formed by the myristic moiety. Therefore, while the residues involved in the stabilization of the protein-membrane complex are essentially the same in both states (Figure 3), the stabilization of the Rec-target complex in the membrane apparently requires persistent electrostatic interactions, specifically salt bridges between the charged guanidinium groups of R43 and R46 and the polar heads of PS lipids. These interactions appear to be necessary to enhance the myristoyl hydrophobic packing in the Rec-GRK1 assembly compared to uncomplexed Rec.

In conclusion, several specific basic residues of Rec (K5, K37 and R43) have been identified by NMR experiments [28] and MD simulations [14] in a non-physiological membrane as well as in agreement with our simulations in a ROS disc-like membrane as drivers of the correct overall orientation of the protein with respect to the membrane. However, other residues identified for the first time in this study, especially R46, seem to play a crucial role for stabilizing both the Rec-R and Rec-GRK1 states by forming persistent salt bridges with PS polar heads. Thus, the specific local membrane composition is indeed a fundamental aspect of the recognition process.

### 2.4. Dynamics of Rec-Bound Biological Membrane 

The importance of specific electrostatic interactions and lipid composition observed in the network of persistent protein-membrane interactions prompted us to further investigate the role of the negatively charged PS head groups in the membrane dynamics following the insertion of Rec-R and Rec-GRK1. The density maps of PS from the upper membrane leaflet as well as those of Rec-R and Rec-GRK1 were compared for each of the five 200 ns MD simulations performed after the completion of the insertion process. The results are shown in Figure 4. Interestingly, the insertion of the myristic group in the membrane (Figure 4A) occurred in the proximity of a region with high density of PS (Figure 4C), confirming that these lipids play an active role in Rec-membrane interaction. This observation holds for both simulated states of Rec (Figure 4B,D). Moreover, once inserted in the membrane, the protein still maintained its contact with PS clusters as shown by density maps obtained from the MD replicas for Rec-R (Appendix A) and Rec-GRK1 (Appendix A), which show a PS enrichment at the membrane-protein interface. Ultimately, the density maps analysis is in good agreement with that concerning the protein-lipid contacts presented in Figure 3, and suggest that PS lipids are necessary to stabilize the electrostatic interactions described in the previous section.

Taken together, these results suggest that, once Rec is sufficiently close to a PS cluster, its basic residues (R43, initially) make contacts with the PS head groups. These contacts seem to hold Rec in place, increasing the time spent by the myristoyl moiety near the membrane surface. This mechanism might enhance the probability for a successful myristoyl insertion. Moreover, once Rec gets attached to the membrane, PS local concentration is enriched as the high number of negatively charged lipids under the protein N-terminal stabilizes its interaction with the membrane.

The formation of lipid clusters should be accompanied by subdiffusive dynamics of specific lipids (PS, PE or PC). To investigate this issue, the motion of lipid molecules over 200 ns-long MD trajectories were analyzed and the lateral mean square displacement (MSD) for each lipid type as a function of time was computed. A direct comparison between the cases where the protein insertion was successful and those where it was not is shown in Appendix A. Although a slight subdiffusive tendency for PS and PC seems to occur after insertion of Rec-GRK1, the evidence is not conclusive. This certainly reflects the anchoring mechanism described in the previous section, although further sampling and larger membrane patches would be necessary to provide quantitative estimates on subdiffusive dynamics.

### 2.5. PSN Analysis of Rec-R and Rec-GRK1 Suggests State-Specific Allosteric Mechanisms 

The structural dynamics of each simulated state of Rec was analyzed by the powerful PSN paradigm, which was recently employed to highlight allosteric properties of three different NCS proteins in a variety of signaling states. The analysis included the assessment of the most robust communication paths between Ca^2+^-binding EF-hands and target interfacial residues for apo Rec (Rec-T), for an intermediate state with one Ca^2+^ ion bound to EF3 (Rec-I) and for the same states analyzed in the present work (Rec-R and Rec-GRK1), though in the absence of the biological membrane [25]. This type of analysis identifies the protein hubs, i.e., the amino acids mediating the highest number of persistent noncovalent interactions (H-bonds, electrostatic and hydrophobic interactions). A comparison between the hubs identified for Rec-R and Rec-GRK1 in the membrane-bound state (Figure 5A,B) shows that some amino acids are essential to mediate intra-molecular interactions in both states. This is the case of D74 and F57 in the N-terminal domain, and D110 in the C-terminal domain, which all maintain essentially the same high connectivity in both states. However, some hubs are apparently specific for the signaling state: K101 and I13 are high-degree hubs only in the absence of the target (Figure 5A) while M132 and K194, both in the C-terminal domain, become highly connected only in the presence of GRK1 (Figure 5B). Hence, the specific signaling states require some rearrangement of the intramolecular connectivity, though without major structural rearrangement.

A major feature of the PSN analysis is that it allows a comparison between each possible intra-molecular route in terms of communication robustness, thus highlighting the shortest and most redundant paths between two selected nodes [24]. This can be extremely useful for NCS proteins to identify the allosteric mechanisms in terms of long-distance connections between the Ca^2+^-binding EF-hand motifs and the interface with the target [25]. In a recent investigation, MD simulations of Rec-R in aqueous solution highlighted a long-distance communication between E121, the bidentate Ca^2+^-binding residue representative for EF3 and F35 in EF1 and Y86 in EF2. The latter residue is involved in the hydrophobic packing of the N-terminal domain. Further, it was reported as a key residue for both intra- and inter-molecular communication with EF3, regardless of the presence of target or Ca^2+^ ions [25]. A similar allosteric path originating from EF3 was observed in the present study, where the robust communication between E121, Y86, F35 and F49 was observed (Figure 5C) when Rec was inserted into the membrane (Rec-R). Notably, while for Rec-R in solution no long-range communication between EF2 (represented by E85) and residues at the C-terminal domain constituting the GRK1 interface was observed [25], the presence of the membrane resulted in a very robust communication between E85 and P190 and Q191, both in EF4 (Figure 5C). 

The presence of the membrane seems to also significantly affect allosteric communication pathways in the Rec-GRK1 case. While EF2 (E85) was only involved in long-range communication with EF4 (E189 and K192) in the absence of the membrane [25], in the present simulation it was still robustly connected with P190 in EF4. However, the most robust communication routes were short-ranged, and involved I52 and F57 in EF2 (Figure 5D). A similar situation stands for E121 in EF3, which was long-distance connected with EF1 (F49) and EF2 (Y86) in the absence of the membrane [25], but was found to be more broadly connected via both long-range (F49 and F57 in EF1) and short-range (Q191 in EF4) communication routes in the presence of the membrane (Figure 5D).

The allosteric effects observed in the present MD simulations for membrane-bound Rec-R and Rec-GRK1 can be explained in terms of changes in intramolecular connectivity with respect to the case where Rec was present in aqueous solution. Figure 6 reports on the change in the degree of each major hub (degree ≥ 7) observed for both Rec-R and Rec-GRK1 when the presence or absence of the biological membrane is concerned.

In general, the presence of the membrane led to a decrease of connectivity of the high-rank hubs for both Rec-R and Rec-GRK1, as highlighted by the almost double number of cases with negative Δ compared to the positive ones (Figure 6). However, it should be noted that the presence of the membrane mostly affects the connectivity of high-degree hubs in the uncomplexed Rec-R state, which showed the highest degree of both positive (K101, ∆ = 4) and negative (F49 and A89, ∆ = −3) variations with respect to the aqueous solution analog (Figure 6). Moreover, the largest variation in hub connectivity for Rec-R mostly involved residues belonging to the N-terminal domain, while it was more prominent in the C-terminal domain for the Rec-GRK1 state.

In conclusion, the analysis of allosteric communications originating from Ca^2+^-binding EF-hands in Rec and ending in amino acids constituting the interface with the GRK1 target suggests an important role for the membrane, that overall permits the establishment of robust longer-range interactions both in Rec-R and Rec-GRK1, which could be fundamental for the recruitment of GRK1 and the stabilization of the supramolecular complex.

## 3. Material and Methods 

### 3.1. Membrane Building and Equilibration

The membrane patch was assembled and hydrated by employing the CHARMM-GUI platform [29] by means of the membrane builder tool [30]. The membrane was constituted by glycerophospholipids with the following composition: 20% of (20:1 ∆^11^)/(20:1 ∆^11^) PS, 40% of (20:1 ∆^11^)/(20:1 ∆^11^) PE and 40% (20:1 ∆^11^)/(20:1 ∆^11^) PC (named by CHARMM-GUI DGPS, DGPE and DGPC, respectively). The final membrane system was composed by 260 lipid molecules and ~64000 atoms in total arranged in a triclinic box with a size of 9 nm along the *x* and *y* axes and almost 4 nm along *z*. The negatively charged membrane patch was automatically neutralized by CHARMM-GUI, using K^+^ and Cl^−^ ions.

All-atom MD simulations were performed using GROMACS [31] 2016.5 package employing a variant of the CHARMM36m force field [32], including the parameters for the myristoylated Gly. 

The system was subjected to energy minimization using the steepest descent algorithm (F_max_ = 1000 kJ·mol^−1^·nm^−1^) and equilibrated at 310 K (Berendsen thermostat [33], coupling constant τ_T_ = 1 ps) for 275 ps of heavy-atom position-restrained MD simulations in canonical (NVT) ensemble. Then, the system was further equilibrated for 10 ns of unrestricted MD simulations in isothermal-isobaric (NPT) ensemble at 310 K (V-rescale thermostat [34], coupling constant τ_T_ = 0.5 ps) and 1 atm (Parrinello-Rahman barostat [35], coupling constant τ_P_ = 5 ps), with semi-isotropic pressure coupling applied separately to lipid and solvent molecules.

The stability of the lipid bilayer throughout the simulation was assessed via a density analysis performed using the *gmx density* function implemented in the GROMACS package. 

### 3.2. Rec Structures

The Rec-GRK1 structure was modelled as elucidated in [25], based on the previous model in [36], while the Rec-R model was obtained by removing the peptide from the complex and by running the same MD simulation protocols as previously described in [37] to allow system relaxation and to avoid artifacts due to peptide removal. The final Rec-R structure after the simulation shows no substantial differences from the Rec-GRK1 starting structure as previously observed experimentally [38]. 

### 3.3. System Assembly and MD Simulations

Rec was placed in a 9 × 9 × 16 nm triclinic box approximately 1 nm above the surface of the bilayer with the major axis perpendicular to the membrane and the N-terminus facing the membrane. The membrane and protein systems were merged and the clashing water molecules at the protein-membrane interface belonging to the original protein box were removed. The system was neutralized with 150 mM KCl and 1 mM MgCl_2_ and subjected to energy minimization first using the steepest descent algorithm (F_max_ = 1000 kJ·mol^−1^·nm^−1^), then with conjugate gradient algorithm (F_max_ = 1000 kJ· mol^−1^·nm^−1)^. The minimized structures were then first equilibrated in the NVT ensemble for 2 ns at 310 K (V-rescale thermostat, τ_T_ = 0.1 ps), then for 2 ns at 310 K and 1 atm in NPT ensemble (V-rescale thermostat, τ_T_ = 0.5 ps, Berendsen barostat, semi-isotropic pressure coupling, τ_P_ = 5 ps), both with harmonic position restraints applied to all heavy atoms (K = 1000 kJ·mol^−1^·nm^−1^). The temperature coupling was separately applied to the protein, lipids and solvent.

The periodic boundary conditions were applied in all directions. A cut-off of 1.2 nm was applied to the Lennard-Jones interactions employing a switching function (switching radius of 10 Å). The electrostatic interactions were treated using the fast smooth particle-mesh ewald (SPME, [39]) with a real-space cutoff of 1.2 nm. The H-bonds atoms were constrained by the LINCS (LINear Constraint Solver) algorithm [40].

Seven 200 ns MD simulations for both Rec-R and Rec-GRK1 states at 310 K and 1.01 atm in NPT ensemble were performed (V-rescale thermostat, τ_T_ = 0.1 ps, Parrinello–Rahman barostat, τ_P_ = 5 ps), two of which started with Rec positioned 1 nm above the membrane. The other five, which differed only in the seed for the randomization of initial velocities, started with Rec anchored to the membrane after myristoyl insertion.

### 3.4. Sampling Homogeneity Checks

Sampling homogeneity was assessed using Principal Component Analysis (PCA) as a criterion for filtering high-frequency motions in MD trajectories. Briefly, the eigenvalues and corresponding eigenvectors of the covariance matrix (C) obtained from the positional fluctuations of Cα [41] were calculated, as elucidated in [24]. The concatenated trajectories were projected onto the first two principal components (PC1 and PC2) identifying the two largest collective motions of the protein. The substantial overlap of the scatter plots in the PC space sampled (Appendix A) is consistent with a homogeneous sampling for each replica.

In addition, the consistency of the conformational sampling was assessed by evaluating the root-mean square inner product (RMSIP) of the essential subspace (ES) of each replica and the concatenated trajectory, calculated as follows:(1)$$RMSIP={\left(\frac{1}{S}\overset{S}{\underset{i=1;\;j=1}{\sum }}{\left(v_{i}^{A}\cdot v_{j}^{B}\right)}^{2}\right)}^{\frac{1}{2}}$$
where v_i_^A^ and v_j_^B^ represent the eigenvectors of the ES belonging to replicas A and B, while S (20 in our case) is dimensionality of the ES.

### 3.5. Lateral Mean Square Displacements

The lateral mean square displacements were calculated via the *gmx msd* function implemented in the GROMACS package. Briefly, the quantity (**r**(t_0_ + t) − **r**(t_0_))^2^ was averaged over all the head groups of lipid molecules of the same type and over multiple time origins for the selected 200 ns-long trajectories.

### 3.6. Analysis of PSN

The dynamic information obtained from the concatenated MD trajectories was then encoded in a static protein structure network (PSN) for each signaling state using PyInteraph [42] software.

Briefly, the software calculates the percentage of frames of MD trajectories in which distance and angle constraints peculiar to each non-bonded interaction (electrostatics, H-bonds and hydrophobic interactions) are fulfilled. An exhaustive list of such constraints is detailed in [24].

The persistence threshold p_T_ was calculated according to the size of the largest hydrophobic cluster criterion [21], resulting in 20.2 for Rec-R and 14.8 for Rec-GRK1. These values were used to filter the three respective interaction graphs, before merging them in a single PSN representing the specific recoverin state.

The intramolecular communication between EF-hands and the Rec-GRK1 interface residues was evaluated using the communication robustness (CR) index, calculated as follows:(2)$$CR_{AB}=\frac{\sigma_{AB}\cdot p_{T}}{l}$$
where p_T_ is the persistence threshold used to filter out non-bonded interactions from the PSN, l represents the length of shortest paths connecting residues A and B, and σ_AB_ is the number of such paths.

### 3.7. Analysis of Protein-Lipid Interactions Persistence

The PSN approach was extended to a protein-membrane structure network to monitor the persistence of the interactions between any lipid in the upper leaflet of the membrane and any protein residue, using an in-house script based on MD Analysis Python library [43,44]. In detail, donor and acceptor atoms, charged atoms and hydrophobic atoms were identified for each lipid molecule type to define intermolecular non-bonded interactions, similarly to the PyInteraph pipeline. The cutoffs used to identify H-bonds, electrostatic and hydrophobic interactions were identical to those used for the PSN analysis.

## 4. Conclusions

Unveiling the detailed structural mechanisms underlying the Rec function may lead to useful generalizations to other NCS proteins, considering the prototypical role of Rec as a myristoyl-switch protein [2]. A number of experimental studies based on NMR [5,28,38,45], surface plasmon resonance [7,12,46,47] and dynamic light scattering [48,49] all confirmed that the binding of two Ca^2+^ ions to the functional motifs EF2 and EF3 in Rec triggers the myristoyl switch process, which by extruding the acyl chain from the protein milieu leads to the significant exposure of a hydrophobic crevice eventually necessary to accommodate the GRK1 target. However, the myristoyl-switch also enhances the affinity of Rec for the membrane, where the acylated GRK1 is located, and the recognition process must thus occur at the water-membrane interface [50]. This study found that the membrane-binding process of Rec is triggered by specific protein-lipid electrostatic interactions, some of which occur prior while others are concomitant with the burial of the myristic moiety. The MD-based network-level analysis of the various Rec states performed in this study suggests that, in order to optimize the sequence of events and eventually stabilize the Rec-GRK1 complex in a peripheral membrane milieu, the system requires specific allosteric interactions between the Ca^2+^- and target-binding regions. These allosteric mechanisms may dynamically change and adapt to the protein state.

## Figures and Tables

**Figure 1 ijms-20-05009-f001:**
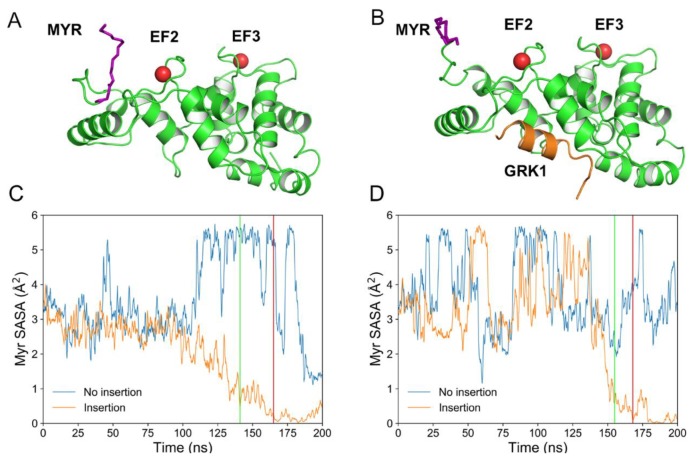
Three-dimensional structures of Rec-R (**A**) and Rec-GRK1 (**B**). Rec (green) and GRK1 peptide (orange) structures are shown as cartoon, Ca^2+^ ions are represented as red spheres, the N-terminal myristoyl group is shown as purple sticks. Time evolution of myristoyl solvent accessible surface area (SASA) over 200 ns replicas computed for Rec-R (**C**) and Rec-GRK1 (**D**). Orange profiles refer to replicas where myristoyl insertion was successful, whereas blue profiles refer to replicas where the insertion failed. The beginning and end of the insertion process are shown by the green and the red lines, respectively. Rec-R insertion starts at 141 ns and terminates at 164 ns (**C**), whereas Rec-GRK1 insertion starts at 153 ns and finishes at 167 ns (**D**).

**Figure 2 ijms-20-05009-f002:**
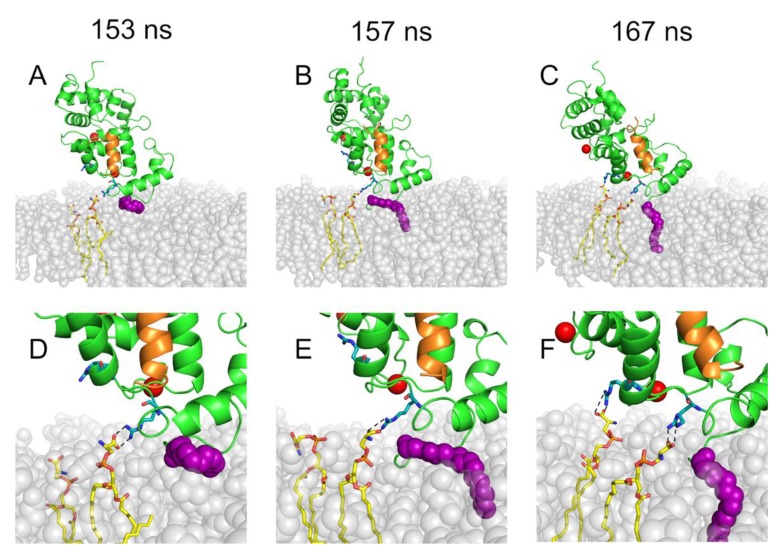
The dynamics of myristoyl insertion into the membrane for Rec-GRK1. Rec (green) and GRK1 peptide (orange) structures are shown as cartoons, Ca^2+^ ions are represented as red spheres, N-terminal myristoyl group is shown as purple spheres, upper leaflet of the membrane is represented as light grey spheres, phosphatidylserine (PS) lipids and Rec residues involved in membrane anchoring are shown as sticks and colored according to chemical elements: O atoms are red, N atoms are blue, P atoms are orange, C atoms are green for Rec and yellow for PS. **A**–**C**: specific steps of insertion referring to the time in the label; **D**–**F**: zoomed-in view of **A**–**C**, respectively.

**Figure 3 ijms-20-05009-f003:**
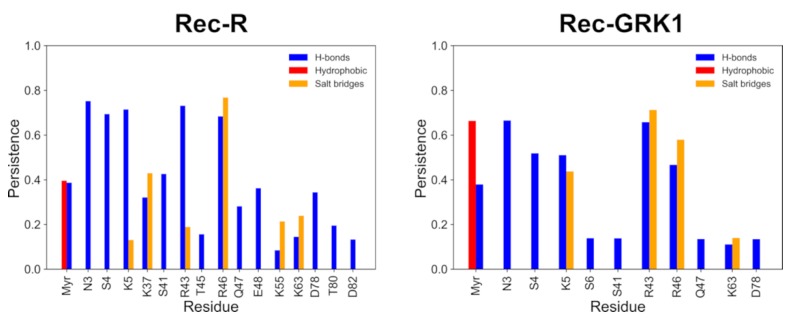
The persistence of the interactions between Rec residues and any lipid in the upper leaflet of the membrane calculated over 1 µs trajectory for Rec-R (**left**) and Rec-GRK1 (**right**). Only residues with a persistence value higher than 0.13 are reported; bars are colored according to the nature of the interactions: H-bonds are shown in blue, hydrophobic interactions in red, salt bridges in yellow.

**Figure 4 ijms-20-05009-f004:**
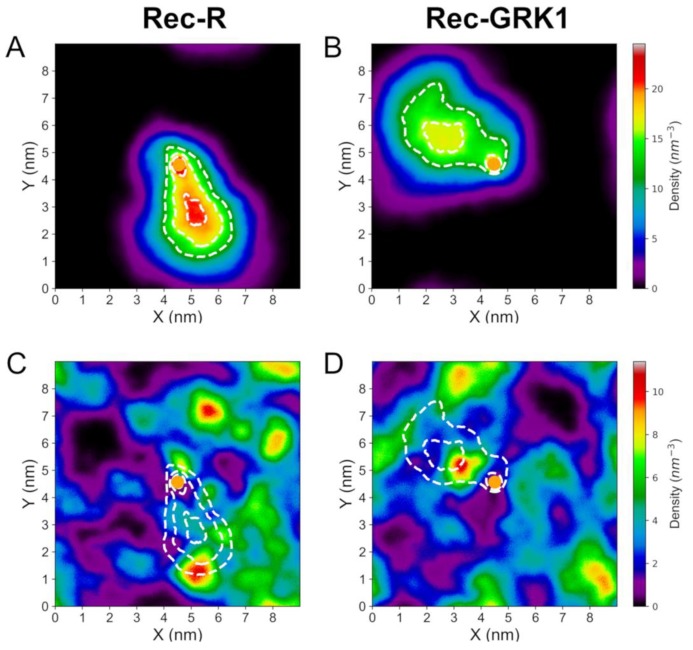
Example of density maps for Rec (**A**,**B**) and PS upper membrane leaflet (**C**,**D**), calculated as the ratio between the number of atoms belonging to each species and the volume. The maps were generated analyzing one 200 ns membrane-anchored trajectory for Rec-R (**A**,**C**) and Rec-GRK1 (**B**,**D**). The trajectories were centered on the myristoyl group, white dashed lines in panels **A** and **B** represent the contour of Rec density profile which is projected onto the PS density maps shown in panels **C** and **D**.

**Figure 5 ijms-20-05009-f005:**
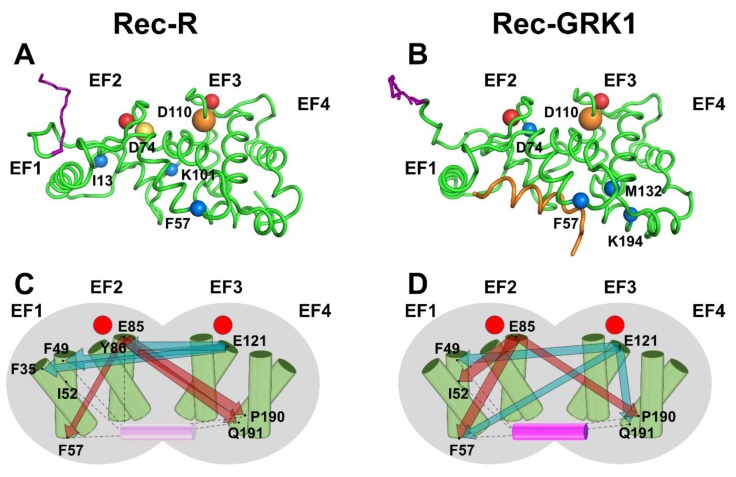
(Top) Three-dimensional structures of Rec-R (**A**) and Rec-GRK1 (**B**). Rec (green) and GRK1 peptide (orange) structures are shown as cartoon, Ca^2+^ ions are represented as red spheres, the N-terminal myristoyl group is shown as purple sticks. Cα atoms of hub residues are shown as spheres with a radius size proportional to their degree (number of connections). The color code is as follows: blue (degree 7), yellow (degree 8) or orange (degree 9). (Bottom) Schematic representation of the intramolecular communication between Rec EF-hands and GRK1 interface residues for Rec-R (**C**) and Rec-GRK1 (**D**) signaling states. Rec shape is represented in grey, the helices of the four EF-hands are represented as green cylinders with the entering helix in front of the exiting helix. GRK1 is shown either as a transparent (**C**) or a solid purple cylinder (**D**) depending on its presence, Ca^2+^-ions are represented as red circles. For both cases, residues E85 and E121, representing EF2 and EF3 respectively, and the three residues with the highest communication robustness (CR) are shown as black dots and labelled. The communication between EF-hands and interface residues is represented by dark red (E85) and teal (E121) arrows whose width is proportional to CR values. Interactions between interface residues and GRK1 peptide are shown by black dashed lines.

**Figure 6 ijms-20-05009-f006:**
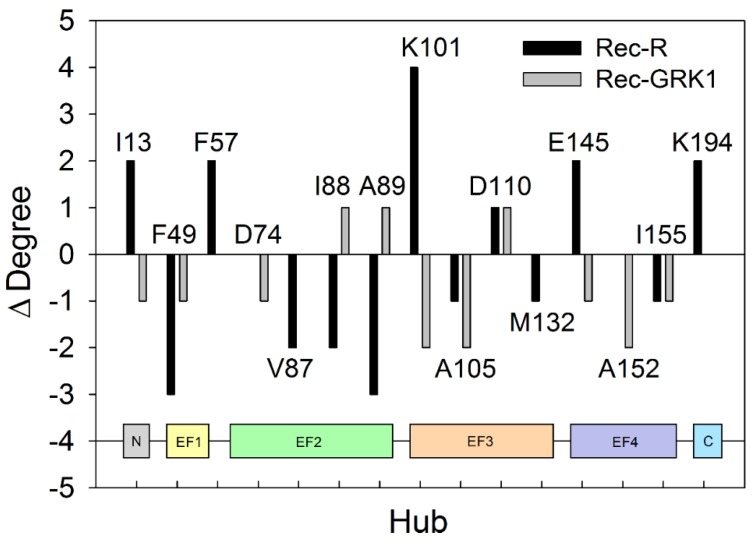
The effects of the membrane on the intramolecular connectivity of Rec-R (black) and Rec-GRK1 (grey). The reported residues were hubs with degree ≥ 7 in one of the four signaling states. Δ Degree was calculated as the difference between the degree of the hubs in the presence and in the absence of membrane. The data referring to the hub degree in the absence of membrane was taken from Supplementary Table T3 in Ref. [25]. The inset represents the structural regions of Rec to which the hub residues belong.

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
