# Peer review of "Effects of Membrane and Biological Target on the Structural and Allosteric Properties of Recoverin: A Computational Approach"

_ijms, 2019, doi:10.3390/ijms20205009_

Round 1
Reviewer 1 Report
Recoverin is a calcium sensor involved in the regulation of phototransduction. Calcium-bound recoverin contains an exposed N-terminal myristoyl group which inserts into the disc membrane of photoreceptor cells and promotes inhibition of the rhodopsin kinase GRK1 by recoverin. In the present manuscript, Borsatto and co-workers investigate the process of insertion of the myristoyl group of recoverin into a phospholipid bilayer by molecular dynamics. They show that insertion is mediated by interactions of recoverin with phosphatidylserine (PS) head groups and that recoverin appears to have a preference for PS-rich regions. They also investigate intramolecular connectivity in recoverin in the presence and absence of membrane and/or a GRK1 peptide, and show that the presence of membrane has a significant effect.
The manuscript is well-written, the data presented in an intelligible manner and appropriately discussed. My only remark regard the title of the manuscript. In my opinion, potential readers expecting an experimental study on "Effects of membrane and biological target on the structural and allosteric properties of recoverin" may be disappointed when discovering that the work is based on computational data. I suggest adding a phrase such as "computational methods", "molecular dynamics" or similar to the title. If it appears too long, the part "a myristoyl-swithc protein" can be omitted without significant impact on the informative nature of the title.
Author Response
We are glad that this Reviewer found merit in our work, and appreciate the suggestion to change the title in order not to create misunderstanding in the reader.
We have followed the Reviewer suggestion and the new title is now:
"Effects of membrane and biological target on the structural and allosteric properties of recoverin: a computational approach".
Reviewer 2 Report
Borsatto et al used ROS disc-like membrane MD simulation found some new crucial Rec-R and Rec-GRK1 states stabilizing residues of Rec, especially the residue R46, which interacts with polar heads of phosphatidylserine lipids. The manuscript is well written. The paper would be great if supported by experimental evidence, like NMR with R46 mutation. Minor suggestion: try to avoid comma in the title.
Author Response
We are glad that this Reviewer found merit in our work. As to the specific comments:
1) "The paper would be great if supported by experimental evidence, like NMR with R46 mutation"
We agree that our findings opened new specific hypotheses, like the one pointed out by the Reviewer, which could be directly tested by focused experiments. However, this falls beyond the scope of our current investigation, and we will certainly consider the suggestion for future studies.
2) "Minor suggestion: try to avoid comma in the title."
We have now changed the title and eliminated the comma.